# Model Comparisons of the Cost Effectiveness of Rubella Vaccination Method in Japanese Adults

**DOI:** 10.3390/vaccines9030233

**Published:** 2021-03-08

**Authors:** Tomoya Itatani, Ryo Horiike, Hisao Nakai, Kazuya Taira, Chika Honda, Fumie Shirai, Kaoru Konishi

**Affiliations:** 1Division of Health Sciences, Doctoral Course of Graduate School of Medical Pharmaceutical and Health Sciences, Kanazawa University, Kanazawa 920-0942, Ishikawa Prefecture, Japan; 2Susaki Regional Welfare and Health Center, Kochi 785-8585, Kochi Prefecture, Japan; spi4949@gmail.com; 3Nursing Department, Kanazawa Medical University, Uchinada 920-0265, Ishikawa Prefecture, Japan; h-nakai@kanazawa-med.ac.jp; 4Department of Human Health Science, Graduate School of Medicine, Kyoto University, Kyoto 606-8501, Kyoto Prefecture, Japan; taira.kazuya.5m@kyoto-u.ac.jp; 5Center for Twin Research, Osaka University Graduate School of Medicine, Suita 565-0871, Osaka Prefecture, Japan; honda-ch@sahs.med.osaka-u.ac.jp; 6Faculty of Nursing, Shiga University of Medical Science, Otsu 520-2192, Shiga Prefecture, Japan; 7Department of Nursing, Morinomiya University of Medical Sciences, Osaka 559-8611, Osaka Prefecture, Japan; fumie_shirai@morinomiya-u.ac.jp; 8Division of Health Sciences, Graduate School of Medicine, Osaka University, Suita 565-0871, Osaka Prefecture, Japan; kkonishi@sahs.med.osaka-u.ac.jp

**Keywords:** cost effectiveness, rubella, congenital rubella syndrome, vaccine

## Abstract

The number of rubella cases has increased in Japan, especially among adults. Rubella infection in pregnant females can lead to congenital rubella syndrome (CRS). The Japanese government is promoting vaccination to prevent CRS. This study performs a cost-effectiveness analysis of the following four methods: (1) females who wished to become pregnant and partners, with an antibody-titer test; (2) females only, with an antibody-titer test; (3) females and males, without an antibody-titer test; (4) females only, without an antibody-titer test. A decision tree model with a hypothetical cohort of 500 males and 500 females was used for the analysis, and the parameters were obtained from previous studies. The number of avoidances of CRS was defined as the effect. Compared to the case where methods were not implemented, the number of CRS cases that can be prevented by implementing the methods was 0.0115589 by (1) and (3) and 0.0147891 by (2) and (4). The cost effectiveness of (1) to (4) was 287,413,677 JPY, 135,050,529 JPY, 388,524,974 JPY, and 197,744,219 JPY, respectively (1 JPY = 0.00963247 USD). Method (2) was the most cost-effective and did not change by sensitivity analysis. We conclude that the vaccination for females only with an antibody-titer test is recommended.

## 1. Introduction

The number of rubella cases is on the rise in Japan, especially among adults. The number of rubella cases reported was 163 in 2015, 126 in 2016, and 91 in 2017, approximately 100 for the third consecutive year. However, the number of reported cases increased significantly to 2946 in 2018 and 2306 in 2019 [1]. The number of reported cases in 2020 was as low as 99 by the 50th week, probably due to the infection prevention measures undertaken throughout the country to curb the new coronavirus epidemic [2]. The number of reported cases of infectious diseases other than rubella is small. For example, as a typical epidemic infectious disease, influenza is prevalent from November to December every year. More than 180,000 cases of influenza were reported in 2019 between 45 and 50 weeks, but only 259 in 2020 [1]. In 2020, as a preventive measure against coronavirus disease 2019 (COVID-19), many people implemented hand hygiene, maintaining social distance, and voluntary quarantine [3]. As a result, it is considered that epidemics of other infectious diseases transmitted by droplets were suppressed in addition to COVID-19. It is unknown how the COVID-19 outbreak and domestic economic activity will coexist. However, if domestic economic activity resumes and people travel back and forth, the number of rubella cases is expected to increase.

The cause of the rubella epidemic in adults lies in past vaccination measures. In Japan, the rubella vaccination started in 1977 and targeted junior high school girls [4]. In 1989, when infants were vaccinated against measles, it was also possible to select the triple vaccine, measles–mumps–rubella (MMR). The MMR vaccine was suspended in 1993 due to the frequent occurrence of aseptic meningitis. However, in 1995, the rubella vaccination for infants was introduced to control the rubella epidemic. Additionally, as a time-limited measure until 2003, junior high school students born between 1979 and 1987 who had never received the rubella or MMR vaccine were vaccinated. However, the vaccination rate was meager at about 50%, and it was feared that a rubella epidemic would occur when this generation becomes adults [5]. In 2013, a rubella epidemic occurred mainly among this generation, and there were more than 14,000 reported cases [6]. This large-scale epidemic in Asia, which was thought to be due to the spread of rubella among unvaccinated people, spread to Japan in 2011 [7]. Many people who contracted rubella in 2019 were in the same generation as in the 2013 epidemic, and most of them were in their 30s and 40s. In particular, there were about four times as many males as females [8]. Under these circumstances, there is a concern that a large-scale rubella epidemic, like 2013, may return.

Rubella is a disease with a good prognosis; however, severe complications may occur in some rare cases. Symptoms tend to become more painful when adults are affected, and it has been reported that about 30% of affected people receive inpatient treatment [9]. The government is promoting vaccination to prevent the disease.

When pregnant women up to the 20th week of pregnancy suffer from rubella, it can lead to congenital rubella syndrome (CRS), which may cause serious complications, including cataracts, sensorineural hearing loss, patent ductus arteriosus, and peripheral arteriopulmonary artery stenosis. CRS incidence varies depending on when the pregnant female is infected with rubella. More than 40 CRS cases were reported during the 2013 epidemic, while there were four reports of CRS in 2019.

The Ministry of Health, Labour and Welfare is promoting vaccination to prevent rubella-infected persons and CRS [10]. This promotion has been notified to the municipalities, and the actual vaccination promotion method is being implemented within the municipal health method framework. Municipalities carry out vaccination promotion methods for adults who wish to become pregnant. The method’s content is a subsidy for vaccination-related costs, including a subsidy for the vaccination and a subsidy for the antibody-titer testing. However, the method of implementation differs depending on the municipality, such as the number of subsidies and the target gender. There are two significant differences in the implementation methods. One is whether to target only females or both sexes. Another is whether to carry out an antibody-titer test before the vaccination. The Ministry of Health, Labour and Welfare recommends vaccination for both adult males and females; however, municipalities need to decide how to implement the methods according to their budgets. Regarding the economic analysis of antibody-titer testing, Ödemiş et al. conducted a cost-effectiveness analysis for nursing students aged 14 to 18 years in Turkey. They stated that it would be cost effective to implement screening tests before the vaccination [11]. Furthermore, Boccalini et al. economically analyzed the appropriate vaccinations for internationally adopted children in Italy. They concluded that antibody testing was more efficacious [12]. Meanwhile, Asli et al. concluded in the study of health care workers and students in Turkey that it is better to get the rubella vaccine without the screening tests [13]. The proportion of rubella-sensitive individuals in the study by Asli et al. was 3.7% for 18–26 year olds and 4.3% for 27–38 year olds, considerably lower than that of Japanese adults. In a Japanese study, Terada et al. analyzed a model in which the rubella vaccine was administered twice, in infancy and in childhood. The study found that urine screening for rubella antibodies before the second vaccination was cost effective [14]. However, the subjects in the study were infants and school children. Currently, Japan’s problem is rubella occurring during adulthood. Therefore, a simulation analysis assuming these are necessary. This study focuses on the subjects and the presence or absence of antibody-titer testing and performs a cost-effectiveness analysis of the following four methods: (1) females who wished to become pregnant (herein, female) and partners (herein, male), with an antibody-titer test; (2) females only, with an antibody-titer test; (3) females and males, without an antibody-titer test; (4) females only, without an antibody-titer test. Based on the results, we will make recommendations on how to implement the methods.

## 2. Materials and Methods

### 2.1. Decision Tree Model and Analysis Method

A cost-effectiveness analysis was performed by simulation using a decision tree. The model is shown in Figure 1. The model predicts the number of rubella patients by defining multiple possible states of the cohort and simulating the cohort transiting dynamics between the states. We constructed four vaccination method models: (1) females who wished to become pregnant and partners, with an antibody-titer test; (2) females only, with an antibody-titer test; (3) females and males, without an antibody-titer test; (4) females only, without an antibody-titer test. Then, we compared the effects of the methods being implemented and not being implemented, respectively.

From the perspective of preventing CRS, a hypothetical cohort of 500 men and women aged 20 to 39 years, a total of 1000 people, was used in the analysis. This value was calculated by dividing the number of births of 1 million in 2013 by the number of municipalities at that time, which was 1742. Thus, the number of births per municipality was approximately 574. To simplify the calculation, we set the virtual cohort for men and women to 500 each. This study targeted males and females who wished to be vaccinated. Therefore, we assumed that all of the targets participated in the vaccination method. The cost was an incremental cost when methods (1) to (4) were implemented compared to when the vaccination-promotion method was not implemented. Incremental costs included vaccination costs, treatment costs in the event of a vaccine side effect, and antibody-titer-testing costs. Medical costs avoided by vaccination were subtracted from the incremental costs. In this study, the number of avoidances of CRS was defined as the effect. For example, the number of CRS occurrences is 0.0155675 when method (1) is not applied, but it drops to 0.0004086 when the method is used. That is, the difference, 0.0151589, is the effect of method (1). The method was evaluated by incremental cost effectiveness. In other words, the cost of preventing one CRS case was described as cost effectiveness, and the economic efficiency of the four methods was compared.

### 2.2. Parameters

The main parameters are listed in Table 1. The incidence rate was estimated based on data from an infectious-disease-outbreak survey [15,16]. From the number of reported rubella cases by age group, rubella’s incidence rate between the ages of 20 and 40 was estimated, and parameters were set for each gender [15]. We obtained population data from e-Stat, which is a portal site for Japanese government statistics [16].

The incidence rate of females (per 100,000):

= (number of infections in those aged 20–40/population aged 20–40) × 100,000

= (1834/14612) × 100,000

≒ 13

Incidence rate in males (per 100,000)

= (number of infections in those aged 20–40/population aged 20–40) × 100,000

= (6239/15146) × 100,000

≒ 42

We estimated the number of CRS cases by multiplying the incidence rate of female rubella cases by the CRS occurrence in each trimester of a rubella-infected pregnant female [17]. CRS due to subclinical infection was also considered during the calculation. According to the Center for Infectious Diseases, between 2002 and 2014, CRS included 44 cases of rubella during pregnancy, compared to 11 cases with no rubella. Therefore, we assumed that subclinical infection would cause 1.25 times the number of rubella infections during pregnancy, which was included in the calculation [18].

The incidence of CRS due to infection in pregnant females:

= (Σ(trimester week × each incidence)/weeks of pregnancy) × 1.25

= ((3 × 1.0) + (4 × 0.83) + (2 × 0.8) + (2 × 0.52) + (2 × 0.45) + (4 × 0.04))/52 × 1.25

≒ 0.2395

Although the initial antibody titer at the vaccination was set at 95% [19], it is reported that acquired immunity with the rubella vaccine would last for life or be maintained for at least 15 years [19]. Therefore, we did not consider the attenuation of immunity obtained by vaccination. In addition, we did not consider the implementation of a booster vaccination. In fact, municipalities in Japan do not subsidize booster vaccinations. Those who are not willing to be vaccinated are not considered to undergo antibody-titer testing. Therefore, we assumed that the subject was vaccinated if the antibody-titer test determined that vaccination was necessary.

When both males and females were vaccinated, we assumed that they were married couples, and the effect of preventing infection between families was considered.

The cost of vaccination and the cost of antibody-titer tests were set regarding medical institutions’ public information. We computed the entire method’s cost by calculating the number of people to be vaccinated × the vaccination rate × the cost for one dose.

We assumed the medical costs for rubella infection as follows. Acute encephalitis and thrombocytopenic purpura have been reported as severe-infected cases of adult complications due to rubella [20]. Inpatients were set at 33.3% of the adult-rubella patients, referring to previous studies [9]. Except for severe cases, there were complications such as fever and arthritis [9]. We assumed that these patients were examined and prescribed in the outpatient department. In setting medical expenses, the hospitalization costs for severe cases were also taken into consideration [20,21]. For each symptom of CRS, we estimated the associated medical expenses from the Survey on Medical Benefit Expenditure and the Patient Survey [21,22].

We assumed the medical costs for the vaccine side effects as follows. Many of the MMR vaccine’s side effects are due to the measles vaccine, while the rubella vaccine is reported to be very safe [19]. Severe adverse events that occur rarely include thrombocytopenia and encephalitis [23]. Further, in this study, the frequency of occurrence of severe cases was set based on a domestic survey [24]. The incidence of severe-vaccine side effects and the medical costs were integrated to calculate the side-effect medical costs converted per person. The medical cost of side effects was added to the cost of the vaccination method.

The medical costs for adult infection and the vaccine side effects include costs due to lost employment opportunities [25], therefore, higher-income males have higher medical costs than females.

### 2.3. Sensitivity Analysis

In a decision-tree-simulation analysis, the results may vary depending on each parameter’s value used in the analysis. Therefore, each parameter’s value was changed to verify the result, and the sensitivity analysis was performed.

All data used in this study were open data and did not contain any personal information.

The cohort is branched because the birth rate differs depending on the age group.

## 3. Results

Comparing with the case where the prevention method was not implemented, the incremental costs for implementing the four vaccination methods (1) to (4) were estimated to be (1) 4,471,785 JPY, (2) 2,075,056 JPY, (3) 6,004,517 JPY, and (4) 3,002,241 JPY, respectively (1 JPY = 0.00963247 USD). The number of affected individuals without the vaccination was estimated to be 0.065 out of 1000 in the cohort, 0.00171 for both males and females vaccinated ((1) and (3)) and 0.00325 for females only ((2) and (4)). The number of CRS outbreaks was estimated to be 0.0155675 if the method was not implemented, 0.0004086 if both males and females were vaccinated ((1) and (3)) and 0.0007784 if only females were vaccinated ((2) and (4)). Therefore, the number of CRS cases that can be prevented by implementing the vaccination method is 0.0151589 for both males and females ((1) and (3)) and 0.0147891 for females only ((2) and (4)). Therefore, the four methods’ cost effectiveness is (1) 287,413,677 JPY, (2) 135,050,529 JPY, (3) 388,524,974 JPY, and (4) 197,744,219 JPY, respectively, and it is estimated that method (2) that conducts antibody-titer testing in advance and vaccinates only females is the most cost-effective (Table 2).

A sensitivity analysis was performed on the parameters of the female incidence rate, the male incidence rate, the male and female incidence rate, the vaccine cost, the antibody-test costs, the medical costs, and the CRS incidence (Table 3). The most sensitive parameter is the incidence rate, and when the incidence rate of females was changed from 10 times to 0.1 times in method (2), the cost effectiveness changed from 8,771,901 JPY to 1,397,836,812 JPY. The next most sensitive parameter was the incidence rate of CRS, and when the CRS incidence rate was changed from 2 to 0.5 times, the cost effectiveness of method (2) changed from 65,240,218 JPY to 274,671,152 JPY. The sensitivity of the other parameters was relatively low and did not significantly affect the results. In the sensitivity analysis of all parameters, method (2) was the most cost-effective and did not change.

## 4. Discussion

In 2013, the population between the ages of 15 and 49, who accounted for most of the rubella epidemic, was 15.5 million females and 16 million males. When the model used in this study was applied to this population, there were 3500 females with rubella and 11,000 males with rubella, for a total of 14,500 people. The actual number of reported cases is 14,348, which is not much different from the model’s estimated value. Additionally, if the number of CRS occurrences estimated by the model is applied to the 2013 population, it will be 31 cases. This value is slightly less than the actual value but does not differ significantly. The decision tree model used this time is judged to be generally valid. Moreover, in the sensitivity analysis, the results were not overturned by the fluctuation of the parameters. Therefore, it is judged that the results of this study are generally reliable.

In the case of the vaccination measures targeting only females, the number of people receiving the vaccination is half that of the vaccination for males and females. Therefore, the incremental cost required for the measure is also reduced to approximately half, considerably suppressed. As for the number of affected people, the total number of virtual cohorts is as small as 1000. Therefore, the absolute number is small for both measures. However, compared to the vaccination cases for both males and females, the number of affected males is 40 times higher when vaccinated using a females-only vaccine.

On the other hand, the number of affected females did not change significantly, regardless of whether males were vaccinated, and increased by about 1.9 times. The incidence of CRS was 1.9 times higher when men were not vaccinated than when they were vaccinated. The cost of antibody testing was reduced when it was performed and was reduced to less than half than when it was not performed. Based on the above, we consider that the antibody test before vaccination should be carried out clearly from an economic point of view. From the viewpoint of controlling the number of infected people, the vaccination for both males and females has a higher control effect. However, from the perspective of suppressing CRS, the vaccination of males is not very useful, and from the viewpoint of cost effectiveness, it is better to vaccinate only females.

To prevent the outbreak of rubella in adults, the vaccination for both males and females reduces the number of infected people. Furthermore, since it is possible to avoid family-to-family transmission, it is desirable to vaccinate both males and females, especially to prevent CRS. On the other hand, expanding the target of vaccination leads to a direct increase in costs. This can be a heavy burden, depending on the economic conditions of each municipality. Some municipalities limit vaccination assistance to a partial burden rather than the full amount. This study suggests that it is more important for females to be vaccinated than for males to control CRS. From this, it is better to provide free vaccination of females and increase females’ vaccination rate, even if the budget for vaccination of males is allocated.

The above is summarized, and the following recommendations are made as follows. In the vaccination method, it is better to perform an antibody test in advance. Both males and females should be vaccinated to control the affected individuals if the municipality has sufficient financial resources. If financial resources are limited, it is better to focus the financial resources on females’ vaccinations to prevent CRS.

### Future Study

The sensitivity analysis revealed that the parameters did not significantly overturn the conclusions. However, to perform a more accurate simulation, it is necessary to improve the parameters’ accuracy. As a result of this study, we suggest that ensuring free vaccination of females and increasing the vaccination rate would lead to CRS’s prevention. For females to be vaccinated reliably, it is necessary not only to make them free of charge but also to encourage them. In Japan, no studies have been conducted to clarify the rate of rubella vaccination in adults. There is a need for future research on the vaccination rate and the related factors to improve the vaccination rate.

According to a study by the National Institute of Infectious Diseases, the rubella antibody kit (Denka Seiken) had a sensitivity of 82.5% to 96.5% and a specificity of 63.2% to 94.3% [26]. If the specificity is low, the individual in need may not be vaccinated, which is not advisable from the perspective of CRS prevention. In addition, if the sensitivity is low, a person who does not require vaccination may be vaccinated, which is economically wasteful. The vaccine-cost-sensitivity analysis in our study did not overturn the results. Therefore, vaccine sensitivity does not affect the results of this study. However, higher sensitivity and specificity are required to carry out a highly accurate economic analysis and CRS prevention.

## 5. Conclusions

Vaccinating both males and females can reduce rubella infections and CRS compared to the vaccination of females only. The most cost-effective method is to carry out an antibody test before vaccination and to vaccinate only females.

Based on the above, we recommend a method of performing antibody testing before vaccination. We recommend that only females be targeted, and if sufficient financial resources are available, we recommend that both males and females be vaccinated.

## Figures and Tables

**Figure 1 vaccines-09-00233-f001:**
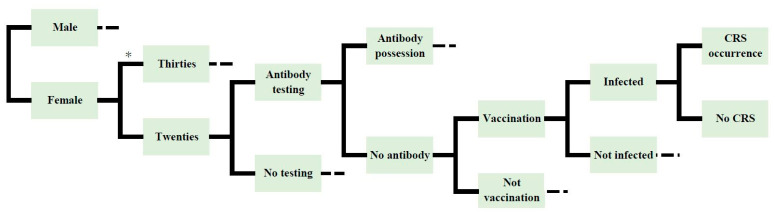
Decision tree model of the vaccination method.

**Table 1 vaccines-09-00233-t001:** Parameters.

Parameters	Value	Unit	Reference
Incidence rate of females	13	per population of 100,000	[15,16]
Incidence rate of males	42	per population of 100,000	[15,16]
Incidence of CRS due to infection in pregnant females	0.2395	%	[17,18]
Initial vaccine efficacy	95.0	%	[19]
Efficacy when vaccinated for both females and males	97.4	%	[19]
Number of female cohorts	500	people	Assumed
Number of male cohorts	500	people	Assumed
Vaccination cost	6000	JPY	Assumed
Medical costs per female infection	165,007	JPY	[9,20,21]
Medical costs per male infection	172,149	JPY	[9,20,21]
Medical costs per CRS	4,570,093	JPY	[21,22]
Medical cost of vaccine side effect of female	4.55	JPY	[23,24,25]
Medical cost of vaccine side effect of male	4.48	JPY	[23,24,25]
Cost of antibody testing	4000	JPY	Assumed

Medical costs include indirect costs due to lost employment opportunities. The cost of vaccination and antibody testing was covered by the local government. JPY = Japanese JPY. 1 JPY = 0.00963247 USD.

**Table 2 vaccines-09-00233-t002:** Cost effectiveness of the four methods.

Vaccination Method		(1)	(2)	(3)	(4)
	No Method Implemented	Both Sex with an Antibody Test	Female Only with an Antibody Test	Both Sex without an Antibody Test	Female Only without an Antibody Test
Number of subject to antibody testing	0	1000	500	0	0
Total cost of antibody test (JPY)	0	4,000,000	2,000,000	0	0
Number of female vaccinated people	0	13	13	500	500
Total cost of vaccinated females (JPY)	0	75,056	75,056	3,002,241	3,002,241
Number of male vaccinated people	0	66	0	500	0
Total cost of vaccinated males (JPY)	0	396,729	0	3,002,276	0
Total vaccination cost (JPY)	0	471,785	75,056	6,004,517	3,002,241
Total antibody testing and vaccine costs (JPY)	0	4,471,785	2,075,056	6,004,517	3,002,241
Number of infected females	0.06500	0.00171	0.00325	0.00171	0.00325
Medical cost for infected females (JPY)	10,725	282	536	282	536
Number of CRS occurrences	0.0155675	0.0004086	0.0007784	0.0004086	0.0007784
Medical cost for CRS (JPY)	71,145	1868	3557	1868	3557
Number of CRS cases prevented	―	0.0151589	0.0147891	0.0151589	0.0147891
Number of infected males	0.21000	0.00551	0.21000	0.00551	0.21000
Medical cost for infected males (JPY)	36,151	949	36,151	949	36,151
Total medical cost for infected people (JPY)	118,022	3098	40,245	3098	40,245
saved medical cost (JPY)	―	114,924	77,777	114,924	77,777
Substantially incremental cost (JPY) (Cost—saved medical cost)	―	4,356,862	1,997,279	5,889,593	2,924,464
Cost-effectiveness (JPY) (Substantial cost/effectiveness)	―	287,413,677	135,050,529	388,524,974	197,744,219
Cost-effectiveness ranking	―	3rd	Most cost-effective	4th	2nd

Medical cost includes infection, congenital rubella syndrome (CRS), and vaccine side effects. JPY = Japanese Yen. 1 JPY = 0.00963247 USD.

**Table 3 vaccines-09-00233-t003:** Sensitivity analysis of the cost effectiveness of the four methods.

Parameters		(1)	(2)	(3)	(4)
		Both Sex with an Antibody Test	Female Only with an Antibody Test	Both Sex without an Antibody Test	Female Only without an Antibody Test
Base case analysis		287,413,677	135,050,529	388,524,974	197,744,219
Incident rate of females	10 times	24,008,216	8,771,901	34,119,346	15,041,270
	0.1 times	2,921,468,291	1,397,836,812	3,932,581,257	2,024,773,707
Incident rate of males	10 times	266,513,622	135,050,529	367,624,919	197,744,219
	0.1 times	289,503,683	135,050,529	390,614,979	197,744,219
Incident rate of sex	10 times	21,918,210	8,771,901	32,029,340	15,041,270
	0.1 times	2,942,368,347	1,397,836,812	3,953,481,312	2,024,773,707
Vaccination cost	9000 JPY	302,963,289	137,586,176	586,429,128	299,170,098
	3000 JPY	271,864,065	132,514,883	190,620,820	96,318,340
Cost of antibody testing	5000 JPY	353,381,729	168,859,156	388,524,974	197,744,219
	3000 JPY	221,445,626	As 101,241,903	388,524,974	197,744,219
Medical cost	2 times	279,832,392	129,791,472	380,943,688	192,485,161
	0.5 times	291,204,320	137,680,058	392,315,617	200,373,748
Incidence of CRS	2 times	141,421,792	65,240,218	191,977,440	96,587,063
	0.5 times	579,397,448	274,671,152	781,620,041	400,058,531

The numerical values in the columns of methods (1) to (4) in the table represent cost effectiveness (substantial cost/effectiveness). Unit = Japanese Yen (JPY). 1 JPY = 0.00963247 USD.

## Data Availability

Publicly available datasets were analyzed in this study. The link of data can be found in “References”.

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
