# Peer review of "Model Comparisons of the Cost Effectiveness of Rubella Vaccination Method in Japanese Adults"

_vaccines, 2021, doi:10.3390/vaccines9030233_

Round 1

Reviewer 1 Report

The current manuscript come across as preliminary study which derives cost effectiveness of rubella vaccine based on simulation analysis. Overall the manuscript is neither presented well nor have provided enough details to be published in current format in Vaccine. Authors did not explain their model  well and neither the underlying disease burden data is presented in clear manner.

1. “ This study performs a cost-effectiveness analysis of the following four methods: (1) females who 21 wished to become pregnant and partners, with antibody titer test, (2) females only, with antibody titer 22 test, (3) females and males, without antibody titer test, (4) only for female.”Sentence not clear.

2. Abstract needs to be simplified. Authors do not provide any information on how they arrive on CRS scores.

3. Revise this sentence “severe problem with rubella is the transmission to pregnant women “

4.“The government is promoting vaccination to prevent rubella-infected persons and CRS” Which Government ? Also correct grammatical mistakes.

5.“There are differences in the implementation methods of the method, depending on the local government” Not sure what authors mean here  ?

6.“Vaccination is an extensive budget method for local governments” Please include numbers ? Vaccination cant be most expensive thing or government ?

7.“The government policy recommends vaccination for both adult males and females, however, the subsidy may be limited to females if a sufficient 90 budget is not available “ What is the source of such information?

8.“Meanwhile, Asli et al. concluded in the study of health care workers and students that it is better to get the rubella vaccine without screening tests. [14]。The proportion of rubella-sensitive individuals in the 98 study by Asli et al. was 3.7% for 18–26 year olds and 4.3% for 27–38 year olds, considerably 99 lower than that of Japanese adults. “ Not clear which country they are referring to. This problem exist for other reference too.

9. What is urinalysis? Also how urinary secretions were used for immune response needs to be elaborated ?

10. “Whether the rubella vaccine should be administered to Japanese adults is not discussed in this study. The recommendation of the rubella vaccine for adults is a national policy.” Sentence are unclear and incoherent.

11. Figure 1 full table should be provided?

12. Methods needs to be elaborate on data and simulation analysis.

13. “The incidence of CRS was set based on previous studies [17,18]. Although the initial antibody titer at vaccination was set at 95% [19],” How vaccination titers were 95% when participation in vaccination drive was low.

14. “Moreover, the cost of treatment for side effects after vaccination was added to the cost of the vaccination method. Acute encephalitis and thrombocytopenic purpura have been 149 reported as severe cases of adult complications due to rubella “Not sure you are talking cost of vaccination or rubella infection here. In case of vaccination side effect, how authors arrive at the numbers.

Author Response

Comments for reviewer 1

Thank you very much for your valuable comment. we have revised the manuscript according to your suggestions. In particular, we added an explanation of the analysis method.

Since “municipalities” and “local governments” were mixed, we unified all expressions as “municipalities.”

  1. > “ This study performs a cost-effectiveness analysis of the following four methods: (1) females who 21 wished to become pregnant and partners, with antibody titer test, (2) females only, with antibody titer 22 test, (3) females and males, without antibody titer test, (4) only for female. ”Sentence not clear.

Response: Thank you very much. The sentence required further elaboration. We added the following sentence to the abstract and line 124 to provide more context: “Females only, without an antibody titer test.”

  1. > Abstract needs to be simplified.

Response: We have simplified the abstract accordingly. The abstract is organized in the order of background, method, result, and conclusion.

>Authors do not provide any information on how they arrive on CRS scores.

Response: We have added how to calculate the number of CRS cases in line 141.

  1. > Revise this sentence “severe problem with rubella is the transmission to pregnant women “

Response: “severe problem with rubella is the transmission to pregnant women.“

Response: This sentence is confusing. Also, the following sentence states that rubella infection in pregnant women leads to CRS. We have deleted this sentence.

  1. > “The government is promoting vaccination to prevent rubella-infected persons and CRS” Which Government ? Also correct grammatical mistakes.

Response: The government being referred to is the “Japanese government,” but to clarify, we have used “The Ministry of Health, Labour and Welfare.”

  1. >“There are differences in the implementation methods of the method, depending on the local government” Not sure what authors mean here  ?

Response: We revised the sentences in line 84. “However, the method of implementation differs depending on the local governments, such as the number of subsidies and the target gender.”

  1. >“Vaccination is an extensive budget method for local governments” Please include numbers ? Vaccination cant be most expensive thing or government ?

Response: The sentences “Financial resources for implementing the method are considered as the reason why the implementation method differs depending on the municipality.” and “Vaccination is an extensive budget method for local governments.” were confusing. The budget allocation of local governments cannot be shown in this study (and the explanation is not suitable for this study). Therefore, we deleted both sentences.

  1. >“The government policy recommends vaccination for both adult males and females, however, the subsidy may be limited to females if a sufficient 90 budget is not available “ What is the source of such information?

Response: We confirmed this directly at the meeting with the staff of the local government. However, such information is not scientifically accurate. Also, there is no previous research. We revised the explanation in line 88 to the following: “The Ministry of Health, Labour and Welfare recommends vaccination for both adult males and females; however, local governments (municipalities) must decide how to implement the methods according to their budgets.”

  1. >“Meanwhile, Asli et al. concluded in the study of health care workers and students that it is better to get the rubella vaccine without screening tests. [14]。The proportion of rubella-sensitive individuals in the 98 study by Asli et al. was 3.7% for 18–26 year olds and 4.3% for 27–38 year olds, considerably 99 lower than that of Japanese adults. “ Not clear which country they are referring to. This problem exist for other reference too.

Response: We have added where the research was conducted. We have also added this information for other studies. Lines 92-96.

  1. >What is urinalysis? Also how urinary secretions were used for immune response needs to be elaborated ?

Response: We changed “urinalysis” to “urine screening for rubella antibodies.” Line 101. We decided the following two sentences were unnecessary and deleted them, “the screening test method was urinalysis,” and “the primary test method used is blood sampling.”

  1. >“Whether the rubella vaccine should be administered to Japanese adults is not discussed in this study. The recommendation of the rubella vaccine for adults is a national policy.” Sentence are unclear and incoherent.

Response: Thank you for pointing this out. These sentences were confusing and unnecessary. We deleted the following sentences: “Whether the rubella vaccine should be administered to Japanese adults is not discussed in this study.” “The recommendation of the rubella vaccine for adults is a national policy. As CRS’s prevention is essential, the probability that the current rubella vaccination promotion method will be discontinued is extremely low.”

  1. >Figure 1 full table should be provided?

Response: The transition of vaccination measures is confusing even for experts living in Japan. We think it is necessary for Japanese researchers. If you strongly suggest deleting it, we shall delete it.

  1. >Methods needs to be elaborate on data and simulation analysis.

Response: We have revised the entire methods section.

  1. > “The incidence of CRS was set based on previous studies [17,18]. Although the initial antibody titer at vaccination was set at 95% [19],” How vaccination titers were 95% when participation in vaccination drive was low.

Response: Do you mean in case of the participation rate of vaccination is low?

Response: On line 128, we added the explanation, “This study targets men and women who wish to be vaccinated. Therefore, we are considering not participating.”

The initial antibody titer when vaccinated is 95%.

Even if the participation rate is considered, the result will not change because the calculation only reduces the number of cohorts.

  1. “Moreover, the cost of treatment for side effects after vaccination was added to the cost of the vaccination method. Acut0e encephalitis and thrombocytopenic purpura have been 149 reported as severe cases of adult complications due to rubella “Not sure you are talking cost of vaccination or rubella infection here. In case of vaccination side effect, how authors arrive at the numbers.

Response: We separated the medical cost of rubella infection from that of vaccine side effects. In addition, a detailed explanation was added in Lines 156-165 and Lines 166-172.

Reviewer 2 Report

Overall this is an interesting study which shows the implications of vaccines being removed from the immunization program in Japan. The authors conduct a cost effectiveness analysis on rubella vaccination programs. Comments as follows:

Overall, you find that the vaccination programs are very expensive, but part of this is that I think you do not include societal costs for CRS?

Other comments:

Could you add US dollars to the abstract?

For (4) in abstract, also mention "without antibody titer test" to distinguish against 2.

In abstract, when you write "The number of CRSs that can be prevented by
25 implementing the four method was 0.0115589 for both males and females, and 0.0147891 for females" - is that for methods 1 and 2 or 3 and 4? More interpretation of these numbers would be helpful. Is this literally saying that every year, by having this vaccination program, on average 0.01 CRSs will be prevented? (seems small)

I would reorganize results to display results for methods 1-4. And specify what your reference group is (I assume no vaccination, or status quo - which is what?).

In Tables 2 and 3 could you write down the units of each measurement?

line 34: I would say rubella is "endemic" more than "prevalent".

Are you considering a 1 dose or 2 dose vaccination strategy?

Are you assuming that everyone tested for rubella would return to get vaccinated?

What is sensitivity/specificity of antibody test?

In Table 3 - second column in rows 2 and 3 you write "method" and "cost-effectiveness" - I think this is confusing because it's not a header for what comes below, so I would just delete it.

Author Response

Comments for reviewer 2

 Thank you very much for your valuable comments. I have revised the manuscript according to your suggestions.

Since "municipalities" and "local governments" were mixed, we rewrote all expressions as "municipalities."

>Overall, you find that the vaccination programs are very expensive, but part of this is that I think you do not include societal costs for CRS?

We have conducted an economic analysis, including the social loss of CRS in the past.

https://www.oatext.com/Cost-benefit-analysis-of-the-rubella-vaccination-in-Japan-to-prevent-congenital-rubella-syndrome-analyses-from-three-perspectives.php#gsc.tab=0

In this analysis, the prevention of CRS itself was set as an "effect," so we decided that it was not appropriate to include the social loss caused by CRS in the analysis. Therefore, this analysis does not consider the social loss of CRS.

Other comments:

  1.  

>Could you add US dollars to the abstract?

Response: We added “1 JPY = 0.00963247” in the abstract. We also inserted US dollar in the main text (line 189) and tables. We changed all Yen to JPY.

2.

>For (4) in Abstract, also mention "without antibody titer test" to distinguish against 2.

Response: Thank you very much. We added "without antibody titer test" to (4) in the Abstract and Methods and Materials. 

>In Abstract, when you write "The number of CRSs that can be prevented by implementing the four method was 0.0115589 for both males and females, and 0.0147891 for females" - is that for methods 1 and 2 or 3 and 4?

>More interpretation of these numbers would be helpful.

Response: Your understanding is correct. Thank you for your suggestion. To make the explanation easier to understand, we added: "(1) and (4)".

>Is this literally saying that every year, by having this vaccination program, on average 0.01 CRSs will be prevented? (seems small)

Response: We added the number of subjects to the abstract.

Response: Due to the low incidence rate of rubella, the incidence of CRS is also low. However, since this economic analysis is a relative comparison of the four vaccine methods, low CRS incidence is not a problem.

 4

>I would reorganize results to display results for methods 1-4. And specify what your reference group is (I assume no vaccination, or status quo - which is what?).

Response: Thank you for your suggestion. We added the sentence "Compared to the case where the methods are not implemented" to the abstract. To make the explanation easier to understand, we added: "(1) and (4)".

5 

>In Tables 2 and 3 could you write down the units of each measurement?

Response: We added the units in Table 2. We also added the unit at the bottom of Table 3.

 6

>line 34: I would say rubella is "endemic" more than "prevalent."

Response: Thank you for your suggestion. We changed the explanation to "The number of rubella cases has increased in Japan" in line 36. We also changed the explanation in the abstract to the same.

>Are you considering a 1 dose or 2 dose vaccination strategy?

 We are considering a 1 dose vaccination strategy.

Response: In line 147, we added the reason why we consider a 1 dose strategy. "In addition, we did not consider the implementation of booster vaccination. In fact, local governments (municipalities) in Japan do not subsidize booster vaccinations."

 7

>Are you assuming that everyone tested for rubella would return to get vaccinated?

Response: Yes, we are assuming that everyone tested for rubella would return.

We added the reason in line 148. "Those who are not willing to be vaccinated are considered to not undergo antibody titer testing. Therefore, we assumed that the subject was vaccinated if the antibody titer test determined that vaccination was necessary."

>What is sensitivity/specificity of antibody test?

Response: The article (web-site) related to sensitivity/specificity of antibody test was found (in Japanese). The publisher is the National Institute of Infectious Diseases. https://www.niid.go.jp/niid/images/idsc/disease/rubella/RubellaHI-EIAtiter_Ver2.pdf

We added the effect of sensitivity/specificity on the vaccine strategy in line 265.

 8

In Table 3 - second column in rows 2 and 3 you write "method" and "cost-effectiveness" - I think this is confusing because it's not a header for what comes below, so I would just delete it.

Response: Thank you for your suggestion. We have deleted it.

Round 2

Reviewer 1 Report

I congratulate the authors for revising the paper. The modelling approach presented here could be important for future decisions. I am still confused at certain places where the writing is still hard to follow and does not make it clear which variables which are coming from previous data and which variables are coming from model. 

There are a few other issues. 

In the method section, the different methods need different headlines,

The title suggests this is analysis of vaccination program while the paper is really about modelling.

Figure 1 is hard to follow.

Author Response

Comments for reviewer 1

Dear reviewer,

Thank you very much for your valuable comment.

I believe that your comment makes our manuscript better.

We have revised the manuscript according to your suggestions.

Tomoya Itatani

Bold is your comment.

>I am still confused at certain places where the writing is still hard to follow and does not make it clear which variables which are coming from previous data and which variables are coming from model.

→Thank you for your suggestion.

We added the description. In particular, incidence rate and the number of CRS cases were explained in detail.

→The document [24] in the text was not included in Table 1, so I added it.

→We deleted “Further, it is said to be about 50% in the first month of pregnancy, 35% in the second month, 18% in the third month, and 8% in the fourth month [10].” in line 76, because detailed data is described in the method section.

→The citations in table1 have also been cleaned up and unnecessary citations have been removed.

>In the method section, the different methods need different headlines,

→Thank you very much.

We made three separated sections.

>The title suggests this is analysis of vaccination program while the paper is really about modelling.

→Thank you very mach.

We changed the title to suggest modelling.

“Cost-Effectiveness Analysis of a Rubella Vaccination Method for Japanese Adults”

“Model Comparisons of the Cost-Effectiveness of Rubella Vaccination Method for Japanese Adults”

>Figure 1 is hard to follow.

→We deleted Figure 1 according to your suggestion. Thank you for your suggestion.